# Interferon-independent STING signaling promotes resistance to HSV-1 in vivo

Lívia H. Yamashiro[1,2], Stephen C. Wilson[1,9], Huntly M. Morrison[1], Vasiliki Karalis[3], Jing-Yi J. Chung[1], Katherine J. Chen[1], Helen S. Bateup [3,4,5], Moriah L. Szpara [6], Angus Y. Lee[7], Jeffery S. Cox[1,8] & Russell E. Vance [1,2,7,8 ✉]

The Stimulator of Interferon Genes (STING) pathway initiates potent immune responses upon recognition of DNA. To initiate signaling, serine 365 (S365) in the C-terminal tail (CTT) of STING is phosphorylated, leading to induction of type I interferons (IFNs). Additionally, evolutionary conserved responses such as autophagy also occur downstream of STING, but their relative importance during in vivo infections remains unclear. Here we report that mice harboring a serine 365-to-alanine (S365A) mutation in STING are unexpectedly resistant to Herpes Simplex Virus (HSV)-1, despite lacking STING-induced type I IFN responses. By contrast, resistance to HSV-1 is abolished in mice lacking the STING CTT, suggesting that the STING CTT initiates protective responses against HSV-1, independently of type I IFNs. Interestingly, we find that STING-induced autophagy is a CTT- and TBK1-dependent but IRF3-independent process that is conserved in the STING S365A mice. Thus, interferon-independent functions of STING mediate STING-dependent antiviral responses in vivo.

[1] Division of Immunology and Pathogenesis, Department of Molecular and Cell Biology, University of California, Berkeley, CA 94720, USA. [2] Howard Hughes Medical Institute, University of California, Berkeley, CA 94720, USA. [3] Division of Neurobiology, Department of Molecular and Cell Biology, University of California, Berkeley, CA 94720, USA. [4] Helen Wills Neuroscience Institute, University of California, Berkeley, CA 94720, USA. [5] Chan Zuckerberg Biohub, San Francisco, CA 94158, USA. [6] Departments of Biology and Biochemistry & Molecular Biology, Center for Infectious Disease Dynamics, Huck Institutes of the Life Sciences, Pennsylvania State University, University Park, Pennsylvania, PA 16801, USA. [7] Cancer Research Laboratory, University of California, Berkeley, CA 94720, USA. [8] Henry Wheeler Center for Emerging and Neglected Diseases, University of California, Berkeley, CA 94720, USA. [9] Present address: Bristol Myers Squibb, 200 Cambridge Park Dr, Cambridge, MA 02140, USA. ✉email: rvance@berkeley.edu

The immune response to pathogens is initiated upon detection of pathogen-associated molecular patterns (PAMPs) such as lipopolysaccharide, flagellin, and nucleic acids[1]. Double-stranded DNA (dsDNA) is an important PAMP for the detection of many pathogens, including *Mycobacterium tuberculosis* and Herpes Simplex Virus-1 (HSV-1)[2–4]. In vertebrates, the intracellular presence of dsDNA is detected by cyclic-GMP-AMP synthase (cGAS), a dsDNA-activated enzyme that produces a cyclic dinucleotide (CDN) second messenger called 2′3′-cyclic-GMP-AMP (2′3′cGAMP)[5–10]. 2′3′cGAMP binds and activates the ER-resident transmembrane protein stimulator of interferon genes (STING). To signal, the C-terminal tail (CTT) of STING recruits TBK1, a kinase that phosphorylates serine 365 (S365) in the CTT[11–14]. Phospho-S365 acts as a docking site for IRF3, a transcription factor that is phosphorylated and activated by TBK1, leading to transcriptional induction of type I interferons (IFNs). Type I IFNs are widely presumed to be the primary output of STING signaling during antiviral defense. However, STING is evolutionarily ancient, and is present even in bacteria[15] and in animals such as the starlet sea anemone *Nematostella vectensis* and *Drosophila melanogaster* that do not appear to encode type I IFNs[16]. In addition to induction of type I IFNs, STING also induces autophagy and NF-κB signaling[17]. These relatively evolutionarily ancient signaling pathways are present in both *N. vectensis* and *D. melanogaster*, raising the possibility that these pathways are the primary or ancestral signaling outputs of STING[18–21].

Here we report the generation of mice harboring a serine 365-to-alanine (S365A) point mutation in STING. These mice are specifically deficient in STING-induced type I IFN responses. STING S365A mice exhibit normal susceptibility to *Mycobacterium tuberculosis* infection but are unexpectedly much more resistant to HSV-1 as compared with STING-null (*Goldenticket*) mice[22]. Likewise, we find *Irf3*−/− mice are relatively resistant to HSV-1. We also report the generation of STING ΔCTT mice. These mice phenocopy the susceptibility of STING null or TBK1 null mice to HSV-1, suggesting that STING mediates protection to HSV-1 via TBK1 recruitment by the STING CTT, independent of STING-dependent IRF3 activation or type I IFN induction. Interestingly, we find that STING-induced autophagy is a TBK1-dependent IRF3-independent process that is present in the STING S365A mice. Our data thus provide evidence for interferon-independent functions of STING that mediate antiviral responses in vivo.

## Results

**STING S365A and ΔCTT mice fail to induce type I IFN.** The relative in vivo importance of the various signaling outputs of STING for antiviral and antibacterial immunity in vertebrates is unknown. To address this issue, we used CRISPR/Cas9 to generate two distinct *Sting* mutant mouse lines: (1) STING S365A mice, which harbor a mutation in *Sting* that results in a serine to alanine substitution at amino acid 365; and (2) STING ΔCTT mice, in which valine 340 has been substituted by a STOP codon, resulting in a STING protein that lacks the entire CTT (Supplementary Fig. 1a, b). We compared the S365A and ΔCTT mice with our previously generated STING-null *Goldenticket* (*Gt*) mice[22]. Since phosphorylation of S365 in the CTT of STING is required for the recruitment and activation of IRF3[11,12,23], we predicted that S365A mice would be deficient in type I IFN responses downstream of STING, but would retain all other STING-dependent signaling events such as autophagy or NF-κB induction. The STING CTT contains S365 and is also essential for recruitment of TBK1[13,14]. Thus, we predicted that ΔCTT mice should also be deficient in all TBK1-dependent responses

downstream of STING. Although the S365A and ΔCTT mutations have been studied previously in vitro, our STING mutant mice provide the first opportunity to address whether the many known in vivo functions of STING[22,24–27] depend on the CTT and S365 phosphorylation.

To test whether endogenous STING requires the CTT and S365 for IFN induction in primary cells, bone marrow-derived macrophages from wild-type (WT) C57BL/6J, *Goldenticket* (*Gt*) STING null mice, and STING S365A and ΔCTT mice were stimulated with STING-specific agonists, including CDNs such as c-di-GMP and 2′3′cGAMP, as well as the cGAS agonist, dsDNA. As controls, cells were also stimulated with Sendai virus (SeV) and poly I:C, which induce type I IFNs via MAVS, independently of cGAS–STING. As expected, stimulation with STING-specific agonists resulted in increased *Ifnb* expression only in WT cells and not in any of the STING mutant cells. By contrast, the IFN response of all four genotypes was similar in response to SeV and poly I:C (Fig. 1a and Supplementary Fig. 1c). STING activation can also lead to production of NF-κB-induced cytokines, such as TNF-α or IL-6[17,28]. Interestingly, primary *Gt*, S365A, and ΔCTT macrophages stimulated in vitro with CDNs or dsDNA were defective for TNF-α induction as compared with WT cells (Supplementary Fig. 1d). However, in vivo stimulation with 5,6-dimethylxanthenone-4-acetic acid (DMXAA), a potent STING agonist[29,30], resulted in measurable TNF-α responses in the serum of WT and STING S365A mice, whereas *Gt* and ΔCTT mice were defective in TNF-α production as expected (Fig. 1b). As a control, the TNF-α response to STING-independent stimuli (e.g., LPS, which activates NF-κB via TLR4) was normal in all genotypes (Fig. 1b). We conclude that S365 may play a role in NF-κB activation, at least in macrophages, but is not required for NF-κB activation in vivo in response to strong STING agonists.

To further characterize our new STING mutant mice, the expression and/or activation of STING and downstream signaling components were assessed by immunoblotting (Fig. 1c). The STING S365A mutation did not affect expression of the STING protein itself or downstream components such as TBK1 and IRF3. STING ΔCTT mice harbor a STING protein of the expected (decreased) molecular weight. Phosphorylation of TBK1—but not of STING or IRF3—occurred in S365A cells in response to STING agonist, consistent with the generally accepted requirement for S365 phosphorylation for IRF3 binding and activation[11,12] (Fig. 1c). By contrast, no phosphorylation of STING, TBK1, or IRF3 was seen in ΔCTT cells, as expected.

**S365A is not required for STING-induced autophagy.** In addition to its role in IFN induction, TBK1 has previously been shown to activate autophagy via the phosphorylation of autophagy adaptor proteins such as NDP52, p62, and optineurin[31]. Likewise, STING activation itself is associated with autophagy-like responses[21,23,32,33]. Interestingly, recent reports claim that STING-induced autophagy does not require the CTT or TBK1[21,34]; however, these experiments utilized conditions that may not reflect the true in vivo requirements, such as over-expressed proteins, immortalized cell lines, and/or artificial in vitro stimulations. In order to investigate whether S365 or the CTT is required for endogenous STING to activate autophagy-like processes, primary macrophages were transfected with 2′3′cGAMP and conversion of LC3-B from form I to the lipidated form II was analyzed. LC3-B conversion was observed in WT and S365A cells in response to 2′3′cGAMP. This response was reduced in *Gt* and ΔCTT cells (Fig. 1d, e, Supplementary Fig. 1e), though quantification of the ratio of LC3II to β-actin band intensities did not reach a significance level of $p < 0.05$. Nevertheless, these results suggested that STING-dependent autophagy

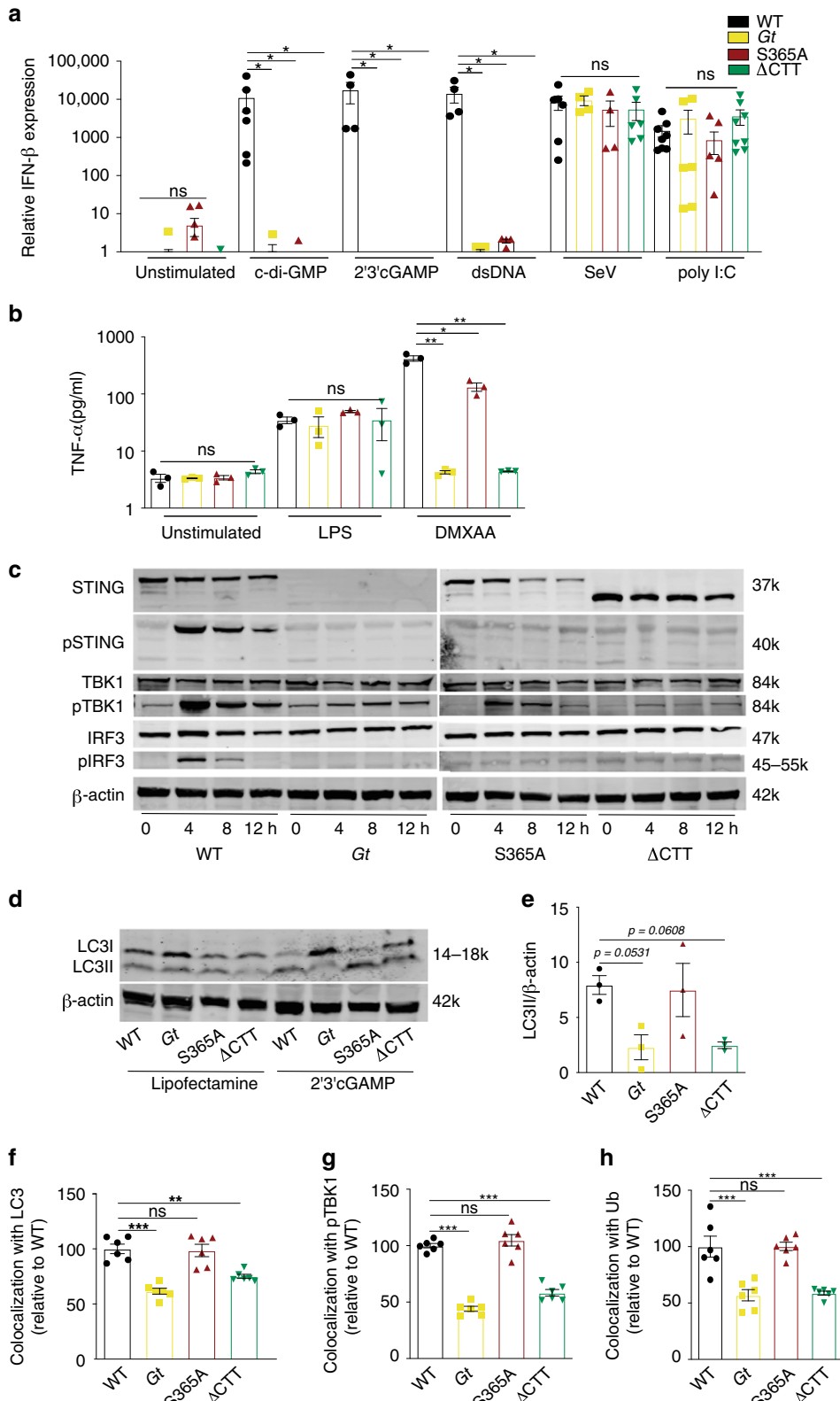

is independent of S365-IRF3 activation and type I IFN responses but largely requires the CTT. To confirm this result using a more quantitative assay, we assessed colocalization of LC3 puncta and cytosolic DNA with an automated image quantitation pipeline. Primary macrophages were transfected for 6 h with Cy3-labeled DNA and colocalization with LC3 puncta was quantified by immunofluorescence. STING-deficient *Gt* and ΔCTT cells

exhibited poor colocalization of DNA and LC3, whereas WT and S365A cells exhibited robust and indistinguishable DNA–LC3 colocalization (Fig. 1f and Supplementary Fig. 1f). Colocalization of DNA with ubiquitin and phospho-TBK1 was also defective in *Gt* and ΔCTT cells but was normal in S365A cells (Fig. 1g, h). STING S365A cells were also able to form puncta in perinuclear regions (Supplementary Fig. 1g). Taken together these data

**Fig. 1 Defective type I IFN induction in STING S365A and ΔCTT macrophages. a** Bone marrow-derived macrophages were stimulated for 6 h and relative expression of IFN-β mRNA was measured. **b** Mice were injected DMXAA (25 mg/kg, i.p.) or LPS (10 ng, i.v.) and TNF-α production was measured on the serum 2 h later (n = 3 mice per genotype). **c** Primary macrophages were transfected with dsDNA for 4, 8 or 12 h or **d** 2′3′cGAMP for 6 h, and cell lysates were analyzed by immunoblotting for the indicated proteins. **e** Quantification of LC3II/β-actin ratio from three independent experiments similar to (**d**). **f** Quantification of LC3, **g** phospho TBK1 or **h** ubiquitin colocalization with DNA in primary macrophages transfected with Cy3-DNA for 6 h. Images were analyzed by an automated pipeline created on Perkin Elmer Harmony software for colocalization quantification (for more details refer to "Methods"). **a** Combined results from two independent experiments. **b**, **c** Representative results of two independent experiments, each yielding similar results. **d**–**h** Representative results of three independent experiments, each yielding similar results. Center and error bars show mean and SEM. Analyzed with one-way ANOVA and Tukey post-test. *p ≤ 0.05; **p ≤ 0.005; ***p ≤ 0.0001. ns, not significant. Exact p values are given in the Supplementary Information.

indicate that endogenous STING requires S365 for IRF3 recruitment and induction of type I IFNs downstream of STING, whereas the CTT (but not S365) is required for TBK1 recruitment and robust autophagy induction. Our new mouse models therefore allow us to genetically separate the IFN- and autophagy-inducing functions of endogenous STING in vivo for the first time.

**No phenotype in *M. tuberculosis*-infected STING mutant mice.** To determine whether STING-induced IFNs and autophagy have distinct functions during in vivo infection, we first examined infections with the bacterium *Mycobacterium tuberculosis*. Previous reports have suggested that the cGAS–STING pathway detects *M. tuberculosis* in macrophages and initiates both a type I IFN response and autophagy-like colocalization of bacteria with LC3[4,35–37]. Type I IFNs exacerbate many bacterial infections, including *M. tuberculosis* infection[38–41], whereas autophagy is generally antibacterial[42]. Therefore, loss of STING may have counteracting effects that obscure its function; indeed, STING-null *Gt* mice do not exhibit dramatic alterations in susceptibility to *M. tuberculosis* infection[36,43]. We hypothesized that perhaps STING S365A mice, which are defective for STING-induced type I IFN induction but not for autophagy, might exhibit enhanced resistance to *M. tuberculosis*. Consistent with this hypothesis, *Irf3*−/− mice have previously been reported to be resistant to *M. tuberculosis*[35]. Therefore, we aerosol infected mice harboring WT, *Gt*, S365A, or ΔCTT STING alleles with virulent *M. tuberculosis*. We found that all STING genotypes were similarly susceptible to *M. tuberculosis* with similar survival rates, bacterial burdens in lungs and spleens, and cytokine production (Fig. 2a–l).

To confirm that STING-induced type I IFN signaling does not affect *M. tuberculosis* susceptibility, we also sought to infect mice lacking the downstream transcription factor, IRF3. However, the published *Irf3*−/− mice that were previously tested are also deficient in *Bcl2l12*, a gene that neighbors *Irf3* and which was inadvertently disrupted by the deletion targeting *Irf3*[44]. Therefore, we generated new *Irf3* deficient (but *Bcl2l12*+/+) mice, as well as *Bcl2l12*−/− (but *Irf3*+/+) mice, using CRISPR–Cas9 (Supplementary Fig. 2a–d). We found *Irf3*−/− mice, *Bcl2l12*−/− mice, and the previously tested doubly deficient mice, were all similarly susceptible to *M. tuberculosis* as WT mice (Supplementary Fig. S2e, f). We cannot explain the previously reported resistance of *Irf3*−/− mice[35] but suspect this may be related to microbiota differences between *Irf3*−/− lines. Nevertheless, we conclude that although *M. tuberculosis* can activate cGAS–STING–IRF3 in macrophages in vitro, STING does not appear to play significant beneficial or detrimental roles in *M. tuberculosis* pathogenesis in vivo.

**STING S365A mice are resistant to systemic HSV-1 infection.** Given that STING is essential for resistance to HSV-1[24,45,46], we next decided to challenge our STING mutant mice with HSV-1. Although induction of type I IFN is presumed to be a major

mechanism of STING-mediated protection against HSV-1, the relative importance of type I IFNs and other STING-dependent responses in host defense against HSV-1 has not been resolved. Indeed, the immune response to HSV-1 is complex and multifactorial. HSV-1 encodes factors to block the type I IFN response, perhaps limiting its effectiveness in control of the infection[45,47,48]. Moreover, it has been shown that neurons do not require type I IFNs—and can instead rely on autophagy—to limit HSV-1 replication in mice in vivo and in vitro[49]. These observations led us to hypothesize that interferon-independent signaling downstream of STING may contribute to control of HSV-1. Initially, mice were intravenously infected with HSV-1 (KOS strain). As expected, WT mice were resistant to infection and remained healthy through 12 days post infection, whereas STING-deficient *Gt* mice were very susceptible to infection and exhibited rapid weight loss and complete paralysis, succumbing 6 days post infection (Fig. 3a–c)[45]. The ΔCTT mice phenocopied the susceptibility of *Gt* mice, demonstrating that the STING CTT is critical for defense against HSV-1. However, in contrast to ΔCTT mice, the S365A mice unexpectedly showed marked resistance to infection, exhibiting only limited weight loss and paralysis, and recovering fully after 6 days of infection (Fig. 3a–c). Susceptibility of *Gt* and ΔCTT mice correlated with elevated viral titers in the brains and spinal cords compared with reduced titers in resistant WT and S365A tissues (Fig. 3d, e). Viral titers among all four genotypes were similarly low in the liver, confirming the neurotropism of HSV-1 (Supplementary Fig. 3a). Given that type I IFNs are essential for resistance to HSV-1[50,51], and that STING is required for type I IFN induction to HSV-1[24,45,46,52], we were surprised that S365A mice were not as susceptible to infection as *Gt* and ΔCTT mice. One possibility to explain this result is that S365A is not required for STING-dependent type I IFN induction in vivo. To test this possibility, we measured expression of *Ifnb* and the interferon stimulated genes *viperin* and *Ifit1* in mice brains following intravenous infection. Only WT brains exhibited a detectable STING-induced IFN response (Fig. 3f, g and Supplementary Fig. 3b). In addition, *Tnf* and *Il6* expression was also elevated only in the brains of WT mice (Supplementary Fig. 3c, d). These data indicate that S365 is critical for STING-induced type I IFN and other cytokines, but surprisingly, this S365-induced response is not critical for STING-dependent resistance to HSV-1.

**STING S365A mice partially resist ocular HSV-1 infection.** HSV-1 is a neurotropic virus that is transmitted via mucosal routes (typically oral, ocular, or genital) and infects epithelial cells before reaching the central nervous system where it establishes latency in neurons[53,54]. Therefore, in order to mimic a more natural route of infection, we challenged mice with HSV-1 using an eye infection model[45,55]. In these experiments, we used strain 17, a more virulent HSV-1 isolate, because the KOS strain used for intravenous infections fails to cause pathology in the eye infection model[55]. As with systemic infection, *Gt* and ΔCTT mice rapidly lost weight and all mice succumbed to infection by

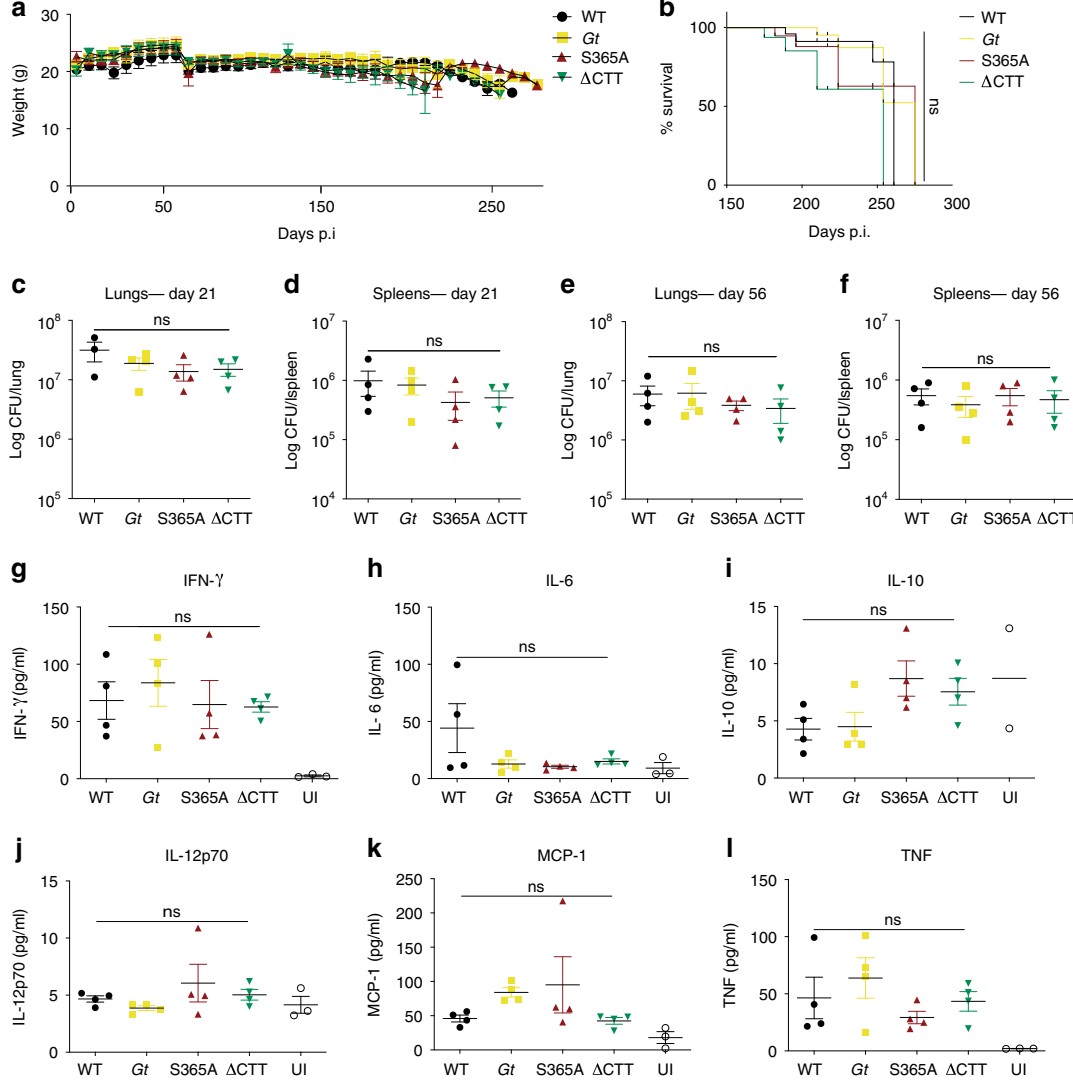

**Fig. 2 STING mutant mice exhibit normal susceptibility to *M. tuberculosis* infection.** Mice were aerosol infected with 400 CFU dose of *M. tuberculosis* (Erdman strain) and **a** weighed every week. **b** Survival of infected mice. **c** Bacterial burden from lungs and **d** spleens at 21 and **e**, **f** 56 days post infection. **g**–**l** Cytokine levels in the lungs from infected mice at day 21, measured by CBA. All mice except C57BL/6J WT were bred in-house. **a**, **b** $n = 12$ mice per genotype. **c**–**f** $n = 4/4/4/4$. **g**–**l** $n = 4/4/4/4/3$. Representative results of five independent experiments, each yielding similar results. UI, uninfected. Center and error bars show mean and SEM. Analyzed with one-way ANOVA and Tukey post test. ns not significant.

6–7 days post infection (Fig. 4a, b). In contrast, WT mice remained fully resistant and S365A mice exhibited an intermediate phenotype, with initial weight loss but later recovery and ~50% survival (Fig. 4a, b). Similar to systemic infection, the susceptibility of the mice correlated with viral burdens: WT and S365A exhibited lower viral titers in the eyewash (Supplementary Fig. 4a), whole brain, brain stem, and spinal cord as compared with *Gt* and ΔCTT mice (Fig. 4c–e). Once again, we found that *Ifnb* and *viperin* expression was elevated in WT but not in *Gt*, S365A or ΔCTT brain stems (Fig. 4f and Supplementary Fig. 4b). By contrast, we did not observe STING-dependent changes in *Tnfa* expression in the eye infection model (Supplementary Fig. 4c)[56]. Previous studies have shown that STING-dependent control of HSV-1 is cell-type specific[45]. To investigate an S365-dependent viral control in brain cells, we infected primary neurons and astrocytes in vitro with HSV-1. However, we observed similar viral yields and autophagy-related processes (colocalization of virus-LC3 and LC3 conversion) (Supplementary Fig. 5a–e) in both cell populations among all genotypes, confirming prior reports that STING does not function cell autonomously in these

cell types[46]. We also infected bone marrow-derived macrophages with HSV-1 and observed that virus does not replicate well in these cells and did not stimulate detectable STING-induced autophagy (Supplementary Fig. 5f, g). To address which cells require S365 for type I IFN induction in vivo, we sorted brain cells (neurons, astrocytes, and microglia) 3 days post infection from brains of HSV-1-infected mice (ocular route). We found elevated *Ifnb* expression in all cell populations only in WT mice (Fig. 4g–i), confirming that IFN-β induction in vivo requires STING S365. Together, our data suggest that STING-mediated control of HSV-1 infection in vivo does not require STING S365-induced type I IFN production.

***Irf3*$^{-/-}$ mice, but not *Tbk1*$^{-/-}$ mice, partially resist HSV-1.** Because TBK1 has been implicated in autophagy induction[4,57,58], whereas IRF3 acts as a transcription factor to induce type I IFNs downstream of STING, we investigated the role of these proteins in the context of an in vivo infection with HSV-1. *Tbk1*$^{-/-}$ mice die as embryos, but this lethality does not occur on a *Tnfr1*$^{-/-}$

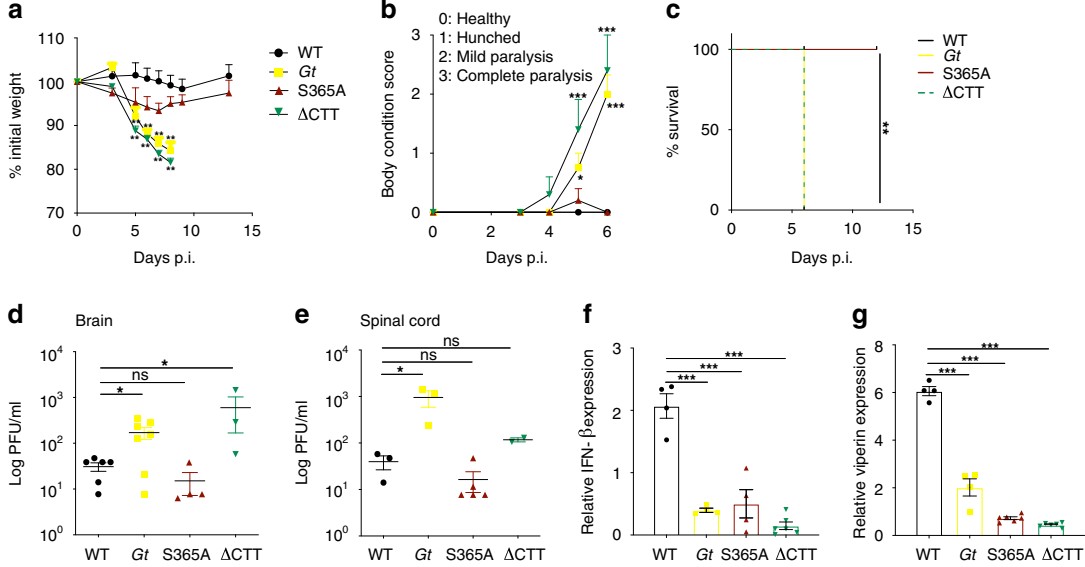

**Fig. 3 S365A mice are partially resistant to systemic HSV-1 infection.** Mice were intravenously infected with $1 \times 10^6$ PFU of HSV-1 (KOS strain). **a** Percentage of initial weight following infection. **b** Body condition score (BCS) of infected mice. **c** Survival of mice following infection. **d** Viral titers in the brain and **e** spinal cord at 6 days p.i. **f** Relative expression of *Ifnb* and **g** *Viperin* in the brain at 3 days p.i. All mice except C57BL/6J WT were bred in-house. **a–c** $n = 7$ mice per genotype, (**d**) $n = 6/7/4/3$, (**e**) $n = 3/3/4/3$, (**f**) $n = 4/3/4/4$, and (**g**) $n = 4/4/6/6$. Representative of at least three independent experiments, each yielding similar results. Center and error bars show mean and SEM. Analyzed with one-way ANOVA and Tukey post test. *$p \leq 0.05$; **$p \leq 0.005$; ***$p \leq 0.0001$. ns not significant. Exact *p* values are given in the Supplementary Information.

background. We therefore analyzed $Tnfr1^{-/-}$ mice compared with $Tbk1^{-/-}Tnfr1^{-/-}$ double deficient mice. $Tbk1^{-/-}Tnfr1^{-/-}$ mice lost weight and succumbed to HSV-1 infection at the same rate as *Gt* and ΔCTT mice, whereas $Tnfr1^{-/-}$ mice were as resistant to HSV-1 as WT mice (Fig. 5a, b). By contrast, $Irf3^{-/-}$ mice presented an intermediate phenotype similar to that of S365A mice. Viral loads in brain stems and total brain correlated with the disease severity (Fig. 5c and Supplementary Fig. 6a) and *Ifnb* expression in the brain stems was increased only in WT mice (Fig. 5d). *Ifnb* expression was also reproducibly decreased in $Tnfr1^{-/-}$ mice in vivo (but not in vitro; Supplementary Fig. 6b) for reasons that are currently unclear. Nevertheless, these results suggest that S365 is critical for STING-induced IRF3 activation and *Ifnb* expression, but neither S365 nor IRF3 are essential for restriction of HSV-1 replication in vivo, whereas the STING CTT and TBK1 are essential.

## Discussion

Our data suggest that the STING CTT and TBK1 can mediate antiviral responses independent of IRF3 activation and type I IFN induction. Although the mechanism that mediates protection to HSV-1 downstream of the STING CTT and TBK1 remains to be elucidated, we propose that a strong candidate is autophagy or an autophagy-like process. Indeed, we found that STING S365A is still able to induce autophagy-like formation of LC3 puncta (Fig. 1d–f), a process previously shown also to require TBK1[32,57]. Autophagy has also previously been shown to be critical for control of HSV-1 and other neurotropic virus infections both in vivo and in vitro[49,59–61]. In fact, HSV-1 has evolved different mechanisms to evade autophagy[60,62,63], but how STING activation initiates autophagy and whether STING-induced autophagy contributes to control of HSV-1 is not clear. In addition, the role of TBK1 in STING-induced autophagy has been a matter of discussion. Some studies show that cells lacking TBK1 can still maintain autophagy-like events (LC3 conversion, puncta formation, and autophagosomes formation)[21,64] while other evidence suggests a critical role

for TBK1 in phosphorylation of selective autophagy receptors and STING autophagosomal degradation[65,66]. Importantly, our data in primary cells suggest that both the CTT and TBK1 are needed for STING-mediated autophagy.

Our data do not prove that the mechanism of STING-induced resistance to HSV-1 requires autophagy. It remains possible that an unidentified CTT–TBK1-induced response (other than, or in addition to, autophagy) is critical for STING-dependent control of HSV-1. Future studies to better elucidate the mechanism of STING-induced autophagy or other STING-induced responses will be required to resolve this uncertainty, as there is no way at present to selectively eliminate STING-induced autophagy (or the putative autophagy-independent CTT–TBK1-dependent process). Nevertheless, our results clearly demonstrate the existence of effective S365/IRF3/interferon-independent antiviral functions for STING.

Type I IFNs are essential for control of HSV-1[45,48,50–52], a result we have confirmed (Supplementary Fig. 6c, d). Thus, our results suggest only that STING-induced IFN, as opposed to all sources of type I IFN, is dispensable for resistance to HSV-1. Although we observe that most type I IFN induction during HSV-1 requires STING (Fig. 4f–i), other pathways for type I IFN induction (particularly the TLR3 pathway)[67–69] have been reported and appear to provide a low but essential type I IFN response.

It remains possible that IRF7, a homolog of IRF3, compensates for the loss of IRF3 and explains the IRF3-independent resistance to HSV-1 that we observe. However, a mechanism for IRF7 activation by STING has not been shown. Our experiments showing that STING S365A mice exhibit resistance to HSV-1 indicate that if IRF7 mediates STING-induced resistance to HSV-1, the mechanism by which IRF7 is activated would have to be independent of S365 phosphorylation and thus distinct from IRF3. Experiments with mice lacking both IRF3 and IRF7 are difficult to interpret because of the essential role of IRF3/7 downstream of other pattern-recognition receptors, e.g., TLR3, that are important for resistance to HSV-1.

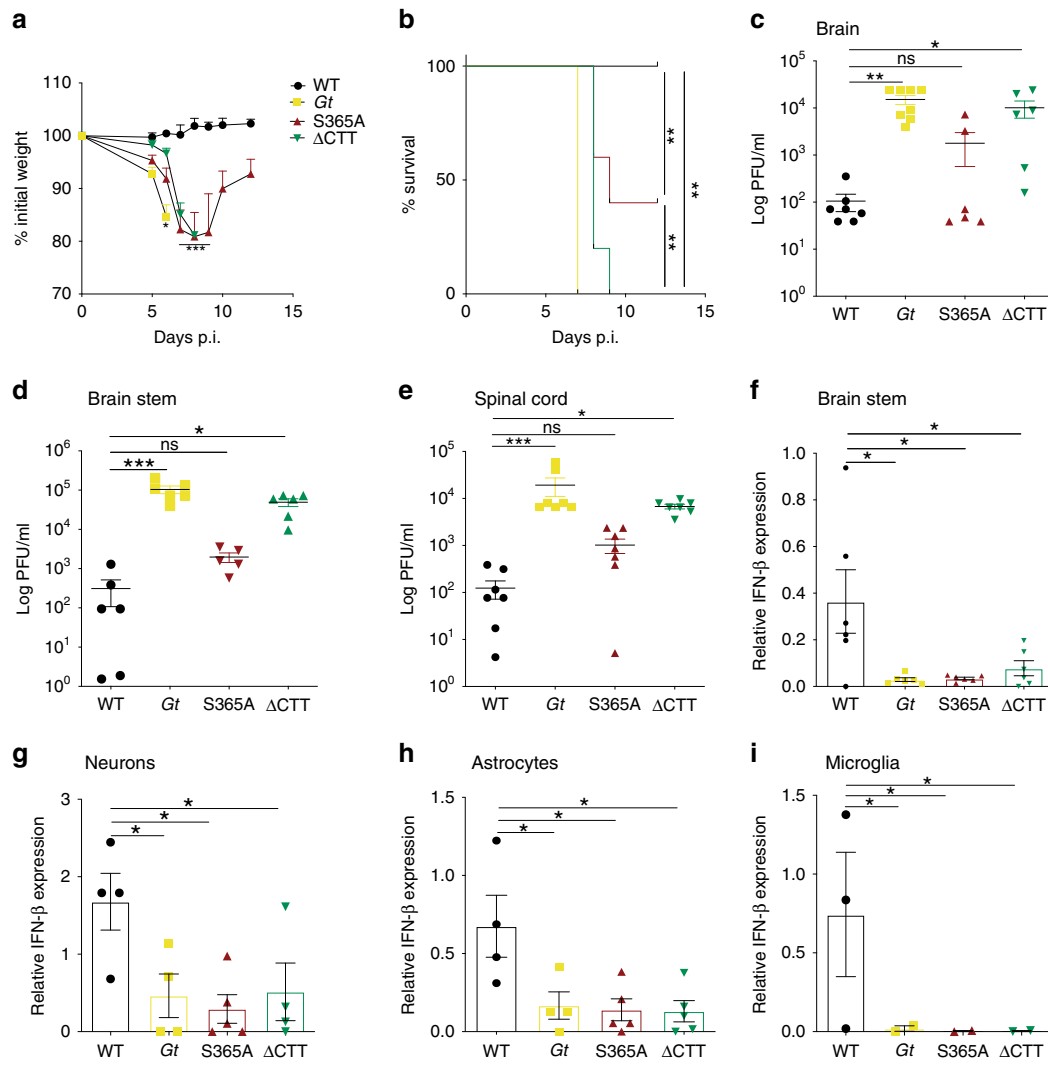

**Fig. 4 S365A mice are resistant to ocular HSV-1 infection.** Mice were infected via the ocular route with $1 \times 10^5$ PFU of HSV-1 (strain 17). **a** Percentage of initial weight following infection. **b** Survival of infected mice. **c** Viral titers in the brain, **d** brain stem, and **e** spinal cord from 6 days p.i. **f** Relative expression of *Ifnb* in the brain stem at 3 days p.i. **g** Brains from infected mice were collected 3 days p.i. and neurons, **h** astrocytes, and **i** microglia cells were sorted, and *Ifnb* expression was analyzed. **a**, **b** n = 7 mice per genotype. **a**, **b** Representative of more than five independent experiments, each yielding similar results. **c–f** n = 3/4 mice per genotype. Combined results from two independent experiments, each yielding similar results. More than five independent experiments were performed. **g**, **h** n = 4 mice per genotype, **i** n = 3 mice per genotype. **g–i** Representative of two independent experiments, each yielding similar results. Center and error bars show mean and SEM. Analyzed with one-way ANOVA and Tukey post-test. *$p \leq 0.05$; **$p \leq 0.005$; ***$p \leq 0.0001$. ns not significant. Exact $p$ values are given in the Supplementary Information.

One interesting feature of our results is that the STING S365A-independent protection we observe is delayed, especially in the eye infection model, and is coincident with the onset of adaptive T-cell responses. Autophagy has been linked to induction of T-cell responses[70–72]. Thus, one attractive possibility is that autophagy is required for antigen processing and presentation to elicit protective adaptive immune responses. Our newly generated STING mutant mice represent valuable tools to dissect this and other putative IFN-independent functions of STING in vivo.

## Methods

**Viruses and reagents.** Dulbecco's Modified Eagle Medium (DMEM) was obtained from Gibco and supplemented with 100 U/ml penicillin, 100 mM streptomycin, and LPS-free FCS (BioWhittaker). DAPI, TRIzol, Poly I:C (all from Invitrogen) Lipofectamine 2000 (Invitrogen) were used in the experiments described below. HSV-1 (strains KOS and strain 17) was grown in Vero cells. Both virus strains were sequence verified by Illumina high-throughput sequencing and de novo assembly[73–75]. The Vero cells used were from the lab stock. The titers of the stocks used were $8–14 \times 10^9$ PFU/ml. Titers were determined by TCID50 assay on Vero cells.

Both strains were used for infection of mice, while only KOS strain was used for in vitro stimulation.

**Mice.** All mice used were specific pathogen free, maintained under a 12 h light–dark cycle (7 a.m. to 7 p.m.), and given a standard chow diet (Harlan irradiated laboratory animal diet) ad libitum. 8–10-week-old male and female mice were used. Mice were kept at temperatures between 68 and 72 F and humidity between 30 and 70%. WT C57BL/6J mice were originally obtained from the Jackson Laboratories (JAX). CRISPR/Cas9 targeting was performed by both pronucleus and cytoplasm injection of Cas9 mRNA, sgRNA, and repair template oligos into fertilized zygotes from C57BL/6J female mice (JAX, stock no. 000664)[76]. STING S365A mice were generated by targeting exon 8 from STING introducing an AGT (serine) to GCC (alanine) substitution at codon 365. The sgRNA sequence was 5′-GCTGATCCATACCACTGATG-3′ and the repair template oligo was C*A*G*ACAAGGCTGTCCCATGCCTCAGATGAGGTCAGTGCGGAGTGGG AGAGGCTGATCCATACCGGCCGATGAGGAGTCTTGGCTCTTGGGACAGTA CGGAGGGAGGAGGTGCCACTGA*G*G*T (underlined is the PAM location). For STING ΔCTT mice, valine 340 was replaced by a premature stop codon. The sgRNA sequence was 3′-GGAGGAAAAGAAGGACTGCT-5′ and the repair template oligo was C*C*C*ACAGACGGAAACAGTTTCTCACTGTCTCAGGA GGTGCTCCGGCACATTCGTCAGGAAGAAAAGGAGGAGTGAACCATGAAT GCCCCCATGACCTCAGTGGCACCTCCTCCCTCC*G*T*A (underlined is the

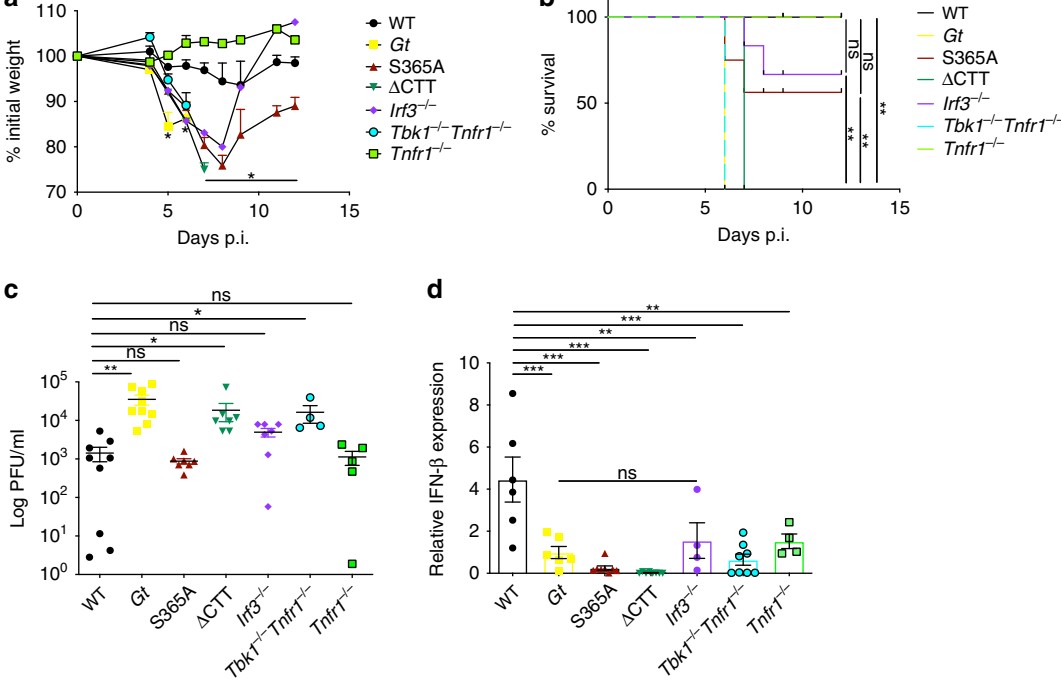

**Fig. 5 STING S365A and *Irf3*−/− mice phenocopy resistance to HSV-1.** Mice were ocular infected with $1 \times 10^5$ PFU of HSV-1 (strain 17). **a** Percentage of initial weight following infection. **b** Survival of infected mice. **c** Viral titers in the brain stem. **d** Relative *Ifnb* expression from brain stems. **a**, **b** $n = 7$ mice per genotype. Representative results of at least two independent experiments, each yielding similar results. **c**, **d** $n = 3/5$ mice per genotype. Combined results from two independent experiments, each yielding similar results. Center and error bars show mean and SEM. Analyzed with one-way ANOVA and Tukey post test. *$p \leq 0.05$; **$p \leq 0.005$; ***$p \leq 0.0001$. ns not significant. Exact $p$ values are given in the Supplementary Information.

PAM location). The asterisks indicate phosphorothioate linkages in the first and last three nucleotides. *Irf3*−/− mice were generated by targeting exon 6 from IRF3. The sgRNA sequence was 5′-GAGGTGACCGCCTTCTACCG-3′. Founder mice were genotyped as described below, and founders carrying mutations were bred one generation to C57BL/6J mice to separate modified haplotypes. Homozygous lines were generated by interbreeding heterozygotes carrying matched haplotypes. *Tbk1*−/−*Tnfr1*−/− and *Tnfr1*−/− mice were described elsewhere[77]. All animal experiments complied with the regulatory standards of, and were approved by, the University of California Berkeley Institutional Animal Care and Use Committee.

**Preparation of gRNA transcript.** DNA oligos (IDT, Coralville, NY) were heated to 95 °C followed by cooling down to room temperature. The self-annealing oligo duplex was cloned into linearized T7 gRNA vector (System Biosciences, Mountain View, CA USA). The cloned sgRNA was sequence verified by DNA sequencing. Then sgRNA template for in vitro transcription (IVT) was prepared by PCR amplification of Phusion high fidelity DNA polymerase (NEB Biolabs, Ipwich, MA), the PCR mixture was cleaned up by PCR cleanup reaction (Qiagen, Hilden, Germany). The sgRNA transcripts were generated by IVT synthesis kit (System Biosciences, Palo Alto, CA). Quality of sgRNA transcripts was analyzed by NanoDrop (Thermo Fisher Scientific, Waltham, MA) and Bioanalyzer instrument (Agilent Technologies, Inc., Santa Clara, CA).

**Genotyping of STING S365A, ΔCTT and *Irf3* alleles.** Exon 8 of STING and Exon 6 of *Irf3* were amplified by PCR using the following primers (all 5′–3′): S365A fwd: CCAACCATTGAAGGAAGGCTCAGTC, S365A rev: CTCACTGTCTCAG-GAGGTGCTCC; ΔCTT fwd: CTAGAGCCCAGACAAGGCTGTCC, ΔCTT rev: CCCACAGACGGAAACAGTTTCTCAC; *Irf3* fwd: AACGTGAGTGCCAGCTGT GG, *Irf3* rev: CTTCACAAGCTTGTCCGTCAGAAACC. Primers were used at 200 nM in a reaction with 2.5 mM MgCl2 and 75 μM dNTPs and 1 U Taq polymerase (Thermo Fisher Scientific) per reaction. Cleaned PCR products were diluted 1:16 and sequenced using Sanger sequencing (Berkeley DNA Sequencing facility).

**Cell culture.** Macrophages were derived from the bone marrow of C57BL/6J or STING mutant (*Gt*, S365A or ΔCTT) mice. Macrophages were derived by 7 days of culture in RPMI 1640 medium supplemented with 10% serum, 100 mM strepto-mycin, 100 U/ml penicillin, 2 mM L-glutamine, and 10% supernatant from 3T3-M-CSF cells, with feeding on day 5. Mouse primary microglia cells and astrocytes were isolated and cultured from the cerebrum of P0 pups. Neonatal cerebra were trypsinized for 20 min and filtered through a 70 μm pore size filter. Cells of three

cerebrum were seeded on one poly-d-lysine-coated 75 cm² culture flask and incubated with DMEM containing 10% FCS. The medium was replaced on day 2 after plating. Henceforth, either microglia or astrocytes were isolated. Astrocytes were isolated using the following method: after 7 days of culture, cells were shaken for 30 min, supernatant was aspirated, and the remaining adherent cells were predominantly astrocytes. Purity of each population was determined by FACS. Primary dissociated hippocampal cultures were prepared from postnatal day 0–1 (P0–1). Mice were euthanized using standard protocols. Bilateral hippocampi from 2 to 3 pups were dissected on ice and pooled together. The tissue was dissociated using 34.4 μg/ml papain in dissociation media (HBSS Ca2+, Mg2+ free, 1 mM sodium pyruvate, 0.1% D-glucose, 10 mM HEPES buffer) and incubated for 3 min at 37 °C. The papain was neutralized by incubation in trypsin inhibitor (1 mg/ml in dissociation media) at 37 °C for 4 min. After incubation, the dissociation media was carefully removed and the tissue was gently triturated, manually, in plating media (MEM, 10% FBS, 0.45% D-Glucose, 1 mM sodium pyruvate, 1 mM L-glutamine). Cell density was counted using a TC10 Automated cell counter (Biorad). For western blot experiments, $2.2–2.5 \times 10^5$ cells were plated onto 24-well plates pre-coated with Poly-D-Lysine (PDL) (Corning) in 500 μl of plating media. After 3 h, plating media was removed and 800 μl maintenance media (Neurobasal media (GIBCO) with 2 mM glutamine, pen/strep, and B-27 supplement (GIBCO)) was added per well. After 4 days in vitro 1 μM cytosine arabinoside (Sigma) was added to prevent glial proliferation. Neurons were maintained in maintenance media for 14 days with partial media changes every 4 days. For immunofluorescence, $2 \times 10^3$ cells were plated in pre-coated 96-well plates (CellCarrier-96 Ultra Microplates, black, Perkin Elmer) following the same procedure.

**Mice *M. tuberculosis* infections.** *M. tuberculosis* strain Erdman (gift of S.A. Stanley) was used for all infections. Frozen stocks of this WT strain were made from a single culture and used for all experiments. Cultures for infection were grown in Middlebrook 7H9 liquid medium supplemented with 10% albumin-dextrose-saline, 0.4% glycerol and 0.05% Tween-80 for 5 days at 37 °C. Mice were aerosol infected using an inhalation exposure system (Glas-Col, Terre Haute, IN). A total of 9 ml of culture was loaded into the nebulizer calibrated to deliver ~400 bacteria per mouse as measured by colony forming units (CFUs) in the lungs 1 day following infection. Mice were sacrificed at various days post infection as indicated in the figure legends to measure CFUs and/or cytokines. All lung lobes were homogenized in PBS plus 0.05% Tween-80 or processed for cytokines (see below), and serial dilutions were plated on 7H11 plates supplemented with 10% oleic acid, albumin, dextrose, catalase (OADC), and 0.5% glycerol. CFUs were counted 21–25 days after plating.

**Cytokine measurements**. Cell-free lung homogenates from *M. tuberculosis*-infected mice were generated[78] by dissociating lungs via passage through 100 μm Falcon cell strainers in sterile PBS with 1% FBS and Pierce Protease Inhibitor EDTA-free (Thermo Fisher). An aliquot was removed for measuring CFU by plating as described above. Cells and debris were then removed by first a low-speed centrifugation (~300 × g) then a high-speed centrifugation (~2000 × g) and the resulting cell-free homogenate was filtered twice with 0.2 μm filters to remove all *M. tuberculosis* for work outside of BSL3. All homogenates were aliquoted, flash-frozen in liquid nitrogen, and stored at −80 °C. Each aliquot was thawed a maximum of twice to avoid potential artifacts due to repeated freeze-thaw cycles. All cytokines were measured using Cytometric Bead Assay (BD Biosciences) according to manufacturer protocols. TNF-α from DMXAA and LPS stimulated mice was also measured by CBA. Results were collected using BD LSRFortessa (BD Biosciences) and analyzed using GraphPad Prism v6.0c. TNF-α from primary macrophages supernatant was measured by ELISA.

**HSV-1 intravenous infection of mice**. Age and sex matched (7–10-week-old) mice were warmed under a lamp for venous dilation and inoculated with $1 \times 10^6$ PFU HSV-1 (KOS strain) in 200 μl of PBS or mock infected with PBS only.

**HSV-1 ocular infection of mice**. Age and sex matched (7–10-week-old) mice, were anaesthetized with intraperitoneal (i.p.) injection of ketamine (100 mg/kg body weight) and xylazine (10 mg/kg body weight). Corneas were scarified using a 25G needle and mice were either inoculated with $1 \times 10^5$ PFU HSV-1 (strain 17) in 5 μl, or mock infected with 5 μl of PBS. Each experiment used 5–10 mice per group. Eyewash was collected by gently proptosing each eye and wiping a sterile cotton swab around the eye in a circular motion. The swabs were placed in 0.5 ml of DMEM medium and stored at −80 °C until the titer was determined. Whole brains, brain stems, spinal cords, and livers were frozen immediately at −80 °C. Tissues were homogenized with tissue homogenizer (Polytron PT 2500 E) for 2 min at frequency 10. Tissues were used for RNA isolation with TRIzol or used for virus titration.

**Scoring and tissue harvest**. Mice were scored for disease, weighed at the indicated times post infection, and euthanized at the specified times post infection for tissue harvesting or once they met end point criteria. The scoring was performed with the following minor modifications[55]: symptoms related to neurological disease named body condition score (0: normal, healthy 1: hunched, 2: uncoordinated, lethargic, mild paralysis, 3: unresponsive/no movement, complete paralysis).

**Sorting of brain cells**. Brains from HSV-1 ocular infected mice (day 3 p.i.) were perfused with 30 ml of PBS (Gibco) and collected on cold HBSS (no Ca$^{2+}$/Mg$^{2+}$) (Gibco) for processing. Neural dissociation kit (Miltenyi Biotec, catalog # 130-092-628) was used for tissue digestion, following manufacture's protocol. Brains were minced with a razor blade and incubated with enzymes mix from the kit. Tissue was dissociated using glass pipettes of different widths (from larger to thinner), filtered using a 70 μm cell strainer, and washed with HBSS (with Ca$^{2+}$/Mg$^{2+}$) (Gibco) to stop enzymes reaction. Digested tissue was proceeded to myelin removal using myelin removal beads (Miltenyi Biotec, catalog # 130-096-731), following manufacture's protocol. Brain suspension was incubated with myelin removal beads and passed through a LS column (Miltenyi Biotec, catalog #130-042-401) in a magnetic field. Cells were washed and prepared for staining. Cells were stained with the following antibodies: APC CD11b (Biolegend, #101212, clone M1/70 dilution 1:100), FITC anti-rat CD90/mouse CD90.1 (Thy-1.1) (Biolegend, #202503, clone OX-7, dilution 1:100), EAAT2/GLT1 (Novus Biologicals, #NBP1-20136SS, dilution 1:100) for 40 min. Secondary staining was performed with donkey anti-rabbit IgG (H + L) PE (eBioscience, #12-4739-81, dilution 1:100). Cells were washed and sorted (for gating strategy, refer to Supplementary Information).

**Infections and cell stimulations**. For infections, bone marrow-derived macrophages from C57BL/6J mice were plated at $1–2 \times 10^6$ cells/well. The next day they were stimulated with CDNs c-di-GMP, 2′3′cGAMP, SeV and poly I:C. Cells were transfected using Lipofectamine 2000 (LF2000; Invitrogen) according to the manufacturer's protocol. All CDNs nucleic acid stimulants were mixed with LF2000 at a ratio of 1 μl LF2000/1 μg nucleic acid, incubated at room temperature for 20–30 min, and added to cells at a final concentration of 4 μg/ml (six-well plates). For SeV, cells were infected at 150 hemagglutination units (HAU)/ml. For poly I:C, 2 mg/ml of the stock solution was heated at 50 °C for 10 min and cooled to room temperature before mixing with LF2000. Transfection experiments were done for 6 h, unless otherwise stated in the figures.

**Immunoblotting**. BMMs were seeded at a density of $1 \times 10^6$ cells per well in six-well tissue culture plates and transfected the next day using Lipofectamine 2000 (Invitrogen) according to the manufacturer's instruction. Cells were lysed at indicated time post transfection with radioimmunoprecipitation assay buffer supplemented with 2 mM NaVO3, 50 mM b-Glycerophosphate, 50 mM NaF, 2 mM PMSF, and Complete Mini EDTA-free Protease Inhibitor (Roche). Proteins separated with denaturing PAGE and transferred to Immobilon-FL PVDF membranes (Millipore). Membranes were blocked with Li-Cor Odyssey blocking buffer.

Primary antibodies were added and incubated overnight. Primary antibodies used were: anti-TBK1 (D1B4) (#3504), anti-phospho-TBK1/NAK (Ser172) (D52C2) (#5483), anti-STING (D2P2F) (#13647), anti-phospho-STING (Ser366) (D7C3S) (#19781), anti-phospho-IRF3 (Ser396) (4D4G) (#4947), anti-LC3B (#2775) all purchased from Cell Signaling Technologies. Anti-IRF3 (EP2419Y) (#ab76409) was from Abcam. Primary antibodies were used at dilution 1:1000. Secondary goat anti-rabbit IgG was conjugated to Alexa Fluor- 680 (Invitrogen) and was used at dilution 1:10,000. Immunoblots were imaged using a Li-Cor fluorimeter. Full blots for all figures presented were provided in the Supplementary Information section.

**Quantitative PCR**. Stimulated cells were overlayed with TRIzol (Invitrogen) and stored. RNA was isolated according to the manufacturer's protocol and was treated with RQ1 RNase-free DNase (Promega). 0.5 μg RNA was reverse transcribed with Superscript III (Invitrogen). SYBRGreen dye (Thermo Fisher Scientific) was used for quantitative PCR assays and analyzed with a real-time PCR system (StepOnePlus; Applied Biosystems). All gene expression values were normalized to *Rps17* (mouse) levels for each sample. The following primer sequences were used: mouse *Ifnb*, (forward) 5′-ATAAGCAGCTCCAGCTCCAA-3′ and (reverse) 5′-CTGTCT GCTGGTGGAGTTCA-3′; mouse *Rps17*, (forward) 5′-CGCCATTATCCCCAGCA AG-3′ and (reverse) 5′-TGTCGGGATCCACCTCAATG-3′; mouse *Viperin*, (forward) 5′-TTGGGCAAGCTTGTGAGATTC-3′ and (reverse) 5′-TGAACCATCTC TCCTGGATAAGG-3′; mouse *Tnf*, (forward) 5′-TCTTCTCATTCCTGCTTG TG G-3′ and (reverse) 5′-GGTCTGGGCCATAGAACTGA-3′; mouse *Il6*, (forward) 5′-GCTACCAAACTGGATATAATCAGGA-3′ and (reverse) 5′-CCAGGTAGCT ATGGTACTCCAGAA-3′.

**Immunofluorescence and high-content imaging**. Bone marrow-derived macrophages were transfected with 0.2 μg of Cy3-labeled DNA for 6 h. Cells were washed with PBS, fixed in 4% paraformaldehyde, and ice-cold methanol. Cells were washed 3× with PBS and blocked and permeabilized with 2% BSA and 0.3% Triton X100. LC3 puncta staining was performed using mouse monoclonal antibody (Nanotools, catalog #0260-100/LC3-2G6 at 1:400, RT) for 3 h, followed by secondary goat anti-mouse IgG labeled with Alexa Fluor 488 (Life Technologies at 1:4000, RT) for 1 h. Nuclei were stained with DAPI. For imaging, cells in 96-well plates were imaged using an Opera Phenix (Perkin Elmer) at RT, using a ×40 1.1 NA water immersion lens (Zeiss). Images were exported to Harmony High-Content Imaging and Analysis Software and automated colocalization measurements were performed with the Perkin Elmer Harmony software version 4.6. package. A pipeline was created to measure colocalization of Cy3-labeled DNA and LC3.

Quantification was performed using data collected from 16 fields per well in 96-well format. Data were then analyzed in Prism using one-way ANOVA analysis. For STING puncta, cells were stained with rabbit polyclonal antibody (Proteintech, catalog # 19851-1-AP) overnight, followed by secondary goat anti-rabbit IgG labeled with Alexa Fluor 547 (Life Technologies at 1:4000, RT for 1 h. Nuclei were stained with DAPI. For imaging, coverslips were imaged using Carl Zeiss LSM710 confocal microscope at RT, using a ×63 oil immersion lens.

**Flow cytometry**. Single suspensions were prepared from each experimental group using a modified protocol as described[79]. Samples were acquired on a FACS X20 Fortessa (BD Bioscience) and analyzed with FlowJo software version 10 (TreeStar).

**Statistical analysis**. All data were analyzed with one-way ANOVA test and Tukey posttest unless otherwise noted and survival data were analyzed with Log-rank (Mantel–Cox) test. Both tests were run using GraphPad Prism version 6. *$p \le 0.05$; **$p \le 0.005$; ***$p \le 0.0001$. All errors bars are SEM and all center bars indicate means. Exact *p* values were provided in the Supplementary Information section.

**Reporting summary**. Further information on research design is available in the Nature Research Reporting Summary linked to this article.

## Data availability
The authors declare that the data supporting the findings of this study are available within the paper and Supplementary Information or from the corresponding author upon request. The source data underlying Figs. 1a, b, e, g, h, 2a–l, 3a–g, 4a–i, 5a–d as well as Supplementary Figs. 1c–e, 2d–f, 3a–d, 4a–c, 5a–d, f, g, and 6a–d are provided as a Source Data file.

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

## Acknowledgements

We thank members of the R.E.V., Barton, Stanley, and Cox labs for discussions, Laura Flores, Peter Dietzen, and Roberto Chavez for technical assistance, Hector Nolla and Alma Valeros and the Cancer Research Laboratory for flow cytometry. We thank Dr. Mary West and Dr. Pingping He of the High-Throughput Screening Facility (HTSF) at UC Berkeley. This work was performed in part in the HTSF, which provided the Perkin Elmer Opera Phenix, funded by NIH Instrument Grant S10OD021828 with the assistance of Christopher Noel. We thank the CNR Biological Imaging Facility at UC Berkeley for the Carl Zeiss LSM710 confocal microscope, funded by NIH S10 program 1S10RR026866-01 with the assistance of Steven Ruzin and Denise Schichnes. We thank Chris Bowen and Daniel Renner in the Szpara lab for assistance with viral genome sequencing and analysis. R.E.V. was supported by an investigator in the Pathogenesis of Infectious Diseases awards from the Burroughs Wellcome Fund. R.E.V. is an HHMI Investigator and is supported by NIH grants AI075039 and AI066302.

## Author contributions

R.E.V. and L.H.Y. designed the experiments. L.H.Y, J.Y.C, and K.J.C. performed the experiments. M.L.S. sequence-verified HSV-1 strains. H.M.M. and J.S.C. helped with autophagy experiments and analysis. V.K. and H.S.B. helped with primary neuronal cultures. R.E.V. and L.H.Y. analyzed the data. S.C.W. and A.Y.L. generated the STING S365A and STING ΔCTT mice. S.C.W. gave technical support and conceptual advice. R.E.V. and L.H.Y. prepared the paper.

## Competing interests

R.E.V. has a financial relationship with Aduro BioTech and Ventus Therapeutics, and both he and the companies may benefit from the commercialization of the results of this research. The other authors declare no competing interests.
