## [Peer Review File · Nature Communications]

Reviewers' comments:

Reviewer #1 (Remarks to the Author):

In this work Tamashiro et al characterize two mouse strains with a STING delta-CTT and S365A mutation, respectively. The authors show that the mice fail to produce IFN β mRNA in response to STING agonists and cGAS-STING-activating pathogens. Despite this, the authors observe only a modest phenotype in HSV1-infected STING S365A mice, which is in contrast to the deltaCTT and STING-deficient mice. Based on this the authors conclude that STING exerts anti-HSV activity independent of type I IFN. By contrast, they observe that autophagy is preserved in the STING S365A mice, based on which (and existing literature) the authors speculate STING utilizes autophagy and ancient IFN-independent antiviral mechanisms to control virus infections in vivo. The authors have generated very strong tools, and have generated some interesting (and potentially provocative) data. However, at the present stage the very bold title is not supported by the data.

1. The main weakness of the work is that the data generated with the pathogenic HSV1 17 strain (which may be the best reflection of human HSV1 infection) show a clear phenotype in the S365A mice (Fig 4), thus arguing directly against the conclusion of the paper.
2. The conclusions are based on the previous report from the Chen lab that S365&366 is the residue responsible for IRF3 recruitment upon phosphorylation. The authors should demonstrate that IRF3 activation is totally abrogated in cells from the mice.
3. Along the same line, can the authors exclude a role for IRF7 in mediating low-grade IFN-induction in the the S365A mice through a mechanism dependent on the CTT, this explaining the phenotypic difference.
4. In many panels, delta-CTT is more impaired than S366A in IFN/ISG induction. Interestingly, this is seen e.g. for the KSV/KOS-induced response (Fig 3f, 5d). Could this contribute to the difference in phenotype?
5. The idea that e.g. autophagy should be responsible for the anti-HSV activity of STING, but should be supported by in vivo data. How does that fit with existing publications concluding that the beclin-binding domain of ICP34.5 has to be deleted to reveal a role for autophagy I anti-HSV defense (Orvedahl et al) .
6. The observation that STING-dependent autophagy is TBK1-independent is surprising and in contrast to the literature (e.g. Gui et al Nature 2019).
7. The IFN/ISG data from the in vivo experiments should be shown together with uninfected controls from the same genotypes. This is critical, since it important to know whether the IFN response is fully abrogated in the different mouse strains
8. Fig 1d. Activation of autophagy should be shown by LC3II/LC3I ratios. In addition, autophagic flux should also be demonstrated (e.g. p62 degradation).
9. The authors should also examine for IFN β protein levels and for IFN α expression (at least at the RNA level)

Reviewer #2 (Remarks to the Author):

The study of Yamashiro et al describes that mice harboring a serine 365-to-alanine (S365A) point mutation in STING are resistant to HSV-1, despite lacking STING-induced type I IFN responses. The authors claim that autophagy and other type I interferon-independent responses mediated by STING are important to confer protection against infection. However, there are no data directly determining the mechanisms of resistance to HSV-1 in these mice. The experiments are well performed and the manuscript is fluent and generally well organized. However, some points mentioned below have to be addressed by the authors before the manuscript can be considered for publication.

- 1) This reviewer thinks that the authors should stress more in the manuscript that this is the first

description of S365A and deltaCTT mice. This would be for their own benefit. Since other investigators could use these animals to test other infection models.

2) I would like to see translocation of STING to perinuclear regions of WT, Gt, S365A and deltaCTT macrophages or fibroblasts transfected with dsDNA as a measure of cell activation. These confocal images would couple perfectly with Figure 1c.

3) Besides LC3, why the authors did not look other autophagy regulators such as Beclin 1 and ULK1 in S365A and deltaCTT cells transfected with cGAMP and dsDNA?

4) This reviewer would like to see a Figure showing autophagy in S365A and deltaCTT mice infected intravenously with HSV-1 in vivo and in macrophages infected with HSV-1.

5) The article of Liu et al (Cell Death Differ. 2019 Sep;26(9):1735-1749) and Gui et al (Nature. 2019 Mar;567(7747):262-266) demonstrated that STING induces non-canonical autophagy that is dependent on ATG5. I suggest to infect ATG5 KO mice with HSV-1 to confirm a susceptible phenotype that could be linked to an autophagy protective mechanism proposed in Yamashiro et al. manuscript.

6) Figure 4d and 4e, explain how many mice were used? Some graphs used 3 other 2 animals. The SE are big because of the few numbers of mice used to determine viral titers.

Reviewer #3 (Remarks to the Author):

Key findings

In the present manuscript Vance and colleagues study the importance of STING-induced type I IFN responses for antimicrobial immunity in vivo. To this end, the investigators generated two new mouse lines: one harbors a STING mutation that renders mice incapable of producing type I IFNs (S365A); another one lacks the C-terminal tail of STING, which is required for recruitment and activation of TBK1, amongst other functions. Using primary cells from these mice, the authors first validate that S365 is essential for STING-dependent type I IFN induction, but not required for promoting autophagy, whereas cells derived from the delta CTT STING mutant mice are defective for both type I IFN induction as well as autophagy induction. The authors then move on to compare the responses of these mice strains and, as a control, the STING-null goldenticket mice, towards infection with *Mycobacterium tuberculosis* and HSV-1, respectively. Whilst there is no difference in the in vivo response towards Mtb infection, the authors show that STING S365A mice do not display an increased susceptibility towards infection with HSV-1 despite majorly compromised type I IFN induction. This result is unexpected, since the general view holds that type I IFN secretion is a major antiviral function of STING.

Overall the study is well-conducted and the results are interesting to a broader community interested in antiviral immunity and the cGAS-STING pathway. Thus, I am in support of publication in Nature Communications. There are a few minor comments, that the authors may want to address before publication.

One minor point relates to the interpretation of the eye infection model, which seems to support an at least partially protective function of STING-mediated cytokine secretion, a point which is not further discussed. Likewise, there appears to be more general role of S365 in regulating cytokine production (including TNF α , IL6) - at least during HSV-1 infection in vivo. Again, a minor aspect that could deserve some space in the discussion.

The main hypothesis put forward by the authors is that STING regulates virus infection through autophagy. With all the mutant mice at hand, one additional experiment that could be added to strengthen this conclusion was to infect BMDMs in vitro with HSV-1 and assess autophagy markers - as performed in Fig. 1 for cGAMP - also, since the authors criticize "artificial in vitro stimulation" for validating STING-dependent autophagy.

Reviewers' comments:

Reviewer #1 (Remarks to the Author):

In this work Yamashiro et al characterize two mouse strains with a STING delta-CTT and S365A mutation, respectively. The authors show that the mice fail to produce IFN β mRNA in response to STING agonists and cGAS-STING-activating pathogens. Despite this, the authors observe only a modest phenotype in HSV1-infected STING S365A mice, which is in contrast to the deltaCTT and STING-deficient mice. Based on this the authors conclude that STING exerts anti-HSV activity independent of type I IFN. By contrast, they observe that autophagy is preserved in the STING S365A mice, based on which (and existing literature) the authors speculate STING utilizes autophagy and ancient IFN-independent antiviral mechanisms to control virus infections *in vivo*. The authors have generated very strong tools, and have generated some interesting (and potentially provocative) data. However, at the present stage the very bold title is not supported by the data.

We thank the reviewer for their generous summary of our manuscript and for their thoughtful and fair critiques. We have included new data and discussed their comments (below).

We also appreciate the reviewer's comment about the title of our manuscript. We recognize that our title ("STING controls Herpes Simplex Virus *in vivo* independent of type I IFN induction") may have confusingly implied that type I IFNs don't matter at all during HSV-1 infection. In fact, our own data (consistent with the literature) confirm that *Ifnar*^{-/-} mice, lacking responses to all type I IFNs, are susceptible to HSV-1. Our original title was merely intended to convey the notion that STING-induced IFN is not strictly required for control of HSV-1, as there appears to be low levels of IFN from other sources that can largely compensate for the loss of STING-induced IFN. The title may have also confusingly implied that STING-induced IFNs do not contribute at all to resistance to HSV-1, which was also not our intent. Indeed, we agree with the reviewers that the STING S365A mice unable to elicit STING-dependent IFN are partially susceptible to HSV-1. The interesting result, in our view, is that these mice are much more resistant than STING null mice, implying that there is an IFN-independent function of STING that is important for resistance to HSV-1. In response to the reviewer's concern about the title, we have changed the title of our manuscript to "An interferon-independent function of STING promotes resistance to HSV-1 *in vivo*". This title avoids the implication that IFNs cannot contribute to resistance to HSV-1, but instead focuses on the main novel finding of the manuscript which is that STING has important IFN-independent functions *in vivo*.

1. The main weakness of the work is that the data generated with the pathogenic HSV1 17 strain (which may be the best reflection of human HSV1 infection) show a clear phenotype in the S365A mice (Fig 4), thus arguing directly against the conclusion of the paper.

Response: We thank the reviewer for this comment, which indicates that we were not as clear as we should have been. As discussed above, we did not mean to imply that S365-induced IFN contributes nothing to resistance to HSV-1. As pointed out by the reviewer, our data clearly indicates the presence of an intermediate (partially susceptible) phenotype in the S365A mice, a result we were not attempting to hide. The important conclusion, however, is that although the levels of IFN in S365A mice are similar to STING null *Gt* mice, they are nevertheless much more resistant to infection than *Gt*. The important result is thus that despite the absence of a substantial contribution of STING to type I IFN responses, STING can still provide substantial resistance *in vivo*. We hope the reviewer agrees that our new title will less confusingly communicate the main take-home message of our paper.

2. The conclusions are based on the previous report from the Chen lab that S365&366 is the

residue responsible for IRF3 recruitment upon phosphorylation. The authors should demonstrate that IRF3 activation is totally abrogated in cells from the mice.

Response: Based on extensive data in the literature from several labs (e.g., Liu et al, 2015; Konno et al, 2013; Tanaka & Chen, 2012), there is a broad consensus in the field that S365 is a crucial site for IRF3 recruitment following STING phosphorylation. However, we agree with the reviewer that it is important to confirm this in our hands using our mice. Our original manuscript did contain the requested data in Figure 1. Specifically, we find that after STING activation, S365A bone marrow macrophages failed to induce IFN- β (phenocopying *Gt* cells) (Fig. 1a) and, in particular, lacked the presence of phospho-IRF3 (Fig. 1c). We presume that the reviewer may have simply overlooked these data, which we acknowledge were only briefly alluded to in the text.

3. Along the same line, can the authors exclude a role for IRF7 in mediating low-grade IFN-induction in the S365A mice through a mechanism dependent on the CTT, this explaining the phenotypic difference.

Response: We agree with the reviewer that IRF7 could theoretically play a compensatory role *in vivo* to induce an IFN response in STING S365A mice. However, we are not aware of any data demonstrating a mechanism for IRF7 recruitment to STING. We assume, given the homology of IRF3 and IRF7, that both transcription factors would be recruited via a similar mechanism — and thus, importantly, the S365A mutation should impair recruitment of both IRF3 and IRF7. Indeed, the amount of IFN induced by S365A mice, *Irf3*^{-/-} mice, and STING null *Gt* mice is indistinguishably low, so we do not find evidence for a compensatory STING-IRF7-IFN response *in vivo*.

However, to further address the reviewer's comment, we performed infections of IRF3/7 double knockout mice. In the event of a compensatory response, one might expect increased susceptibility in the double KO mice as compared to IRF3 single KO or S365A mice. We performed two experiments (see figure for reviewers below), each of which gave slightly different results. Experiment #1 showed similar susceptibility of *Irf3*^{-/-}, *Irf3/7*^{-/-} and S365A mice. Experiment #2 was a lot more noisy, though both the *Irf3*^{-/-} and *Irf3/7*^{-/-} mice trended toward increased susceptibility to HSV-1 infection. It is difficult to know how to interpret these results, but one take-home message is that *Irf3*^{-/-} and the double *Irf3/7*^{-/-} mice behaved similarly to each other, so again, we do not find strong evidence of a substantial compensatory role for IRF7.

Overall, however, it is important to emphasize that experiments with IRF3/7 DKO mice are very difficult to interpret because we know that IRF3/7 play important roles in other IFN-inducing pathways (independent and in addition to their potential roles downstream of STING). Some of these pathways, e.g., the TLR3 pathway, may provide a small amount of IFN that is essential to control HSV-1. Indeed, given that IFNAR KO mice are susceptible to HSV-1, it would not be surprising if IRF3/7 DKO mice were also susceptible. This result would in no way undermine our main conclusion that STING-S365-induced IFN is not essential for STING-dependent resistance to HSV-1 *in vivo*.

Figure 1 for reviewers. Mice were infected via the ocular route with 1×10^5 PFU of HSV-1 (strain 17). Viral titers in the brain stem from day 6. Error bars are SEM. Analyzed with t test. ** and *** $p \leq 0.05$. ns, not significant.

4. In many panels, delta-CTT is more impaired than S366A in IFN/ISG induction. Interestingly, this is seen e.g. for the HSV/KOS-induced response (Fig 3f, 5d). Could this contribute to the difference in phenotype?

Response: We appreciate the reviewer's close attention to our results. The slightly increased IFN seen in S365A mice compared to Δ CTT mice in these panels is not significantly different and is not a consistent finding. In other occasions (e.g. Fig. 4f, 4g) we observed the opposite, i.e., 'higher' IFN- β expression in the Δ CTT cells, or even in *Gt* cells (Fig. 3g). Despite this, the phenotype remains the same (S365A mice are resistant, Δ CTT mice are susceptible). In our experience, even a sensitive assay such as quantitative RT PCR is not reliable for assessing differences when the levels of transcript are barely detectable, as is the case in all the STING mutant mice we analyzed. The apparent differences observed are likely just attributable to the noise of the assay at these very low levels of transcript induction. Thus, we do not make much of the differences between STING S365A, Δ CTT and *Gt* mice. The important take-home point and only reliable finding is that all of these mice have much lower levels of IFN induction compared to WT mice.

5. The idea that e.g. autophagy should be responsible for the anti-HSV activity of STING but should be supported by *in vivo* data. How does that fit with existing publications concluding that the beclin-binding domain of ICP34.5 has to be deleted to reveal a role for autophagy in anti-HSV defense (Orvedahl et al).

Response: We agree with the reviewer that our manuscript does not provide direct *in vivo* evidence that STING-induced autophagy protects against HSV-1 (nor do we claim so in the manuscript). As we point out in our manuscript, since there is no way to specifically eliminate STING-induced autophagy, it is impossible for anyone to address this point at present. We tried hard to assess colocalization between HSV-1 and autophagy markers in neurons after *in vivo* infection, but this experiment proved technically impossible due to the relatively small number of infected neurons and the difficulty in imaging HSV-1 and autophagy markers in these cells. However, despite this, our data are certainly consistent with the idea that STING-induced autophagy restricts HSV-1. This idea is particularly attractive because of *in vivo* data from the Iwasaki group (Yordy et al., 2012) using *Atg5*-floxed mice that shows autophagy—or at least, an *Atg5*-dependent process—does restrict HSV-1 in neurons. It

should be noted, however, that the Yordy paper also found that Atg5 functions to restrict even WT HSV-1 with an intact ICP34.5 gene. In addition, analysis of the “bbd” mutant, that harbors a specific mutation in ICP34.5 that prevents autophagy inhibition, revealed that this mutant is attenuated even in Atg5-deficient mice. These results imply (a) that the bbd mutant is not specifically defective in autophagy avoidance, but is more generally attenuated, and (b) that even wild-type HSV-1 is restricted by Atg5 (“autophagy”). To confirm these general observations in our hands, we performed *in vivo* infections with HSV-1 bbd mutant. We found that this mutant was severely attenuated in all genotypes (see below). These results are consistent with Yordy paper and lead us to conclude the bbd mutant is neuroattenuated, likely for reasons other than a failure to avoid autophagy or even STING.

Figure 2 for reviewers. Mice were ocular infected with 1×10^5 PFU of HSV-1 or Δ BBD HSV-1 strains. Viral titers in the brain stem from day 6. Error bars are SEM. Analyzed with one-way ANOVA and Tukey post-test. **, $p \leq 0.005$. ns, not significant.

6. The observation that STING-dependent autophagy is TBK1-independent is surprising and in contrast to the literature (e.g. Gui et al Nature 2019).

Response: We assume the reviewer means that it is surprising that STING-induced autophagy is TBK1-dependent (not independent), as this is what we show in the paper and is in contradiction to the results of Gui et al., 2019. However, we were not surprised that our results differ from Gui et al. This paper was based solely on overexpression in immortalized cell lines and our results are based on *in vivo* analysis of knock-in mutations at the endogenous *Sting* locus. Our data do suggest that there may be a small amount of CTT/TBK1-independent autophagy induction (e.g., our results show in Fig. 1d that in the Δ CTT cells there is residual presence of LC3 conversion, an autophagy read out). This weak response may dominate when STING is overexpressed, and thus may explain the results of Gui et al. In addition, as we note in the paper, our results showing a TBK1-dependent autophagy induction downstream of STING are not particularly surprising. One of the major known activities of TBK1 is to promote autophagy via the phosphorylation of autophagy adaptors. If anything, in our view it is the claim of Gui et al that STING-induced autophagy is TBK1-independent that is surprising.

7. The IFN/ISG data from the *in vivo* experiments should be shown together with uninfected controls from the same genotypes. This is critical, since it important to know whether the IFN response is fully abrogated in the different mouse strains.

Response: The focus of our paper is that there is a STING-mediated but IFN-independent response that contributes to resistance to HSV-1. To establish this point, our view is that the

useful comparison is between STING null (*Gt*) mice and STING S365A mice. We observe indistinguishable induction of IFN by these strains, yet only the STING S365A mice show resistance. We do not claim that there is no interferon induction in STING null or STING S365A mice. In fact, we believe that there must be at least some low but protective IFN induction via a STING-independent pathway (given that IFNAR KO mice are still susceptible- Suppl. Fig. S6c and S6d). Therefore, we do not expect to find IFN levels in the S365A mice as reduced as uninfected controls. We believe it is sufficient to show that levels are reduced at similar levels as *Gt* mice. In any case, since we have only limited numbers of mice, we never include an uninfected control group in our experiments. To include such a group would require us to go back and repeat all the experiments in the paper, which we are (literally) unable to do.

8. Fig 1d. Activation of autophagy should be shown by LC3II/LC3I ratios. In addition, autophagic flux should also be demonstrated (e.g. p62 degradation).

Response: We appreciate and concur with the reviewer's comment. Indeed, LC3II/LC3I ratio was shown in the originally submitted version of the manuscript as supplemental data on Suppl. Fig. S1d. This result indicates that, as expected, the LC3II/I ratio in S365A is similar to WT and greater than *Gt*.

We have tried to address autophagic flux by p62 degradation but were not successful. BMMs were transfected with 2'3'cGAMP for 8h total and protein expression was measured by immunoblotting. Two hours after stimulation, cells were treated with bafilomycin A to inhibit final degradation of autophagic cargo while others were left untreated to allow complete process and full p62 degradation as controls. However, as shown below we failed to detect autophagic flux as there was no difference in p62 expression. We believe p62 may not be a good indicator for autophagy processes in primary macrophages, as colocalization with labeled DNA also did not show a phenotype among the different cell genotypes. We believe that our existing data, showing STING-dependent effects on LC3 colocalization with DNA, and LC3 conversion, are sufficient to establish the main claims of the manuscript.

b

Figure 3 for reviewers. a, BMMs were stimulated with 2'3'cGAMP for 8h. Two hours after stimulation cells were treated with bafilomycin A or left untreated. Cells lysates were analyzed by immunoblotting for the indicated proteins. **b,** Quantification of p62-DNA colocalization in primary macrophages transfected with Cy3-DNA for 6h. Images were analyzed by an automated pipeline created on Perkin Elmer Harmony software for colocalization quantification. Error bars are SEM. Analyzed with one-way ANOVA and Tukey post-test. ns, not significant.

9. The authors should also examine for IFN β protein levels and for IFN α expression (at least at the RNA level)

Response: To address the reviewer's request, we have tried measuring IFN- β from infected mice but the kit was not sensitive enough. We measured IFN- α *in vitro* in macrophages transfected with 2'3'cGAMP and observed similar results with increased expression only in WT cells.

Figure 4 for reviewers. BMMs were transfected with 2'3'cGAMP for 4h and relative expression of IFN- α mRNA was measured. Error bars are SEM. Analyzed with one-way ANOVA and Tukey post-test. **, $p \leq 0.005$; ***, $p \leq 0.0001$.

Reviewer #2 (Remarks to the Author):

The study of Yamashiro et al describes that mice harboring a serine 365-to-alanine (S365A)

point mutation in STING are resistant to HSV-1, despite lacking STING-induced type I IFN responses. The authors claim that autophagy and other type I interferon-independent responses mediated by STING are important to confer protection against infection. However, there are no data directly determining the mechanisms of resistance to HSV-1 in these mice. The experiments are well performed and the manuscript is fluent and generally well organized. However, some points mentioned below have to be addressed by the authors before the manuscript can be considered for publication.

Response: We thank the reviewer for their positive evaluation of our manuscript. We agree with the reviewer that we do not provide direct evidence for the *in vivo* mechanism of resistance to HSV-1 in the S365A mice. Our data are consistent with the hypothesis that autophagy mediates resistance since autophagy is already known to confer protection to HSV-1, and since we observe intact autophagy responses in the S365A mice. Of course, it is always possible that there are additional mechanisms of resistance. This would always be true and is difficult to rule out. Nevertheless, we feel that our finding that IFN induction by STING does not appear to be required for resistance to HSV-1 to be a very surprising result that is of great importance to the field (and that others will be interested to follow up in future studies).

1) This reviewer thinks that the authors should stress more in the manuscript that this is the first description of S365A and deltaCTT mice. This would be for their own benefit. Since other investigators could use these animals to test other infection models.

Response: We thank the reviewer for this comment and have included a sentence (line 30) addressing the importance of these mice in other infection and tumor models: "Although the S365A and Δ CTT mutations have been studied previously in *in vitro*, these mice provide the first opportunity to study whether the many known functions of STING^{1-3,5,6} depends on the CTT and S365 phosphorylation."

2) I would like to see translocation of STING to perinuclear regions of WT, Gt, S365A and deltaCTT macrophages or fibroblasts transfected with dsDNA as a measure of cell activation. These confocal images would couple perfectly with Figure 1c.

Response: We thank the reviewer for this excellent suggestion. We have completed the requested studies and now include images of BMMs from WT and STING mutant mice transfected with 2'3'cGAMP showing STING puncta in perinuclear regions (New Figure S1f), also reprinted below for reviewers.

Figure 5 for reviewers (also now new Fig. S1f): STING puncta formation is increased in WT and S365A cells. Fluorescence images of primary macrophages transfected for 4h with 2'3'cGAMP. Images were taken using a Carl Zeiss LSM710 confocal microscope.

3) Besides LC3, why the authors did not look other autophagy regulators such as Beclin 1 and ULK1 in S365A and deltaCTT cells transfected with cGAMP and dsDNA?

Response: We also thank the reviewer for this excellent suggestion. As requested, we checked colocalization of dsDNA with ubiquitin and phospho TBK1 as read-outs for autophagy-related responses. These data are shown below in Figure 6 for reviewers and are included in the revised manuscript as Fig. 1g and 1h. We were not successful in developing quantitative and reliable assays to measure Beclin-1 and ULK1.

Figure 6 for reviewers (also now Fig. 1g and 1h). Quantification of colocalization of Cy3-DNA with **a**, phospho TBK1 or **b**, ubiquitin in primary macrophages transfected with Cy3-DNA for 6h. Images were analyzed by an automated pipeline created on Perkin Elmer Harmony software for colocalization quantification. Error bars are SEM. Analyzed with one-way ANOVA and Tukey post-test. ***, $p \leq 0.0001$. ns, not significant.

4) This reviewer would like to see a Figure showing autophagy in S365A and deltaCTT mice infected intravenously with HSV-1 *in vivo* and in macrophages infected with HSV-1.

Response: We tried hard to assess colocalization between HSV-1 and autophagy markers in neurons after *in vivo* infection, but this experiment proved technically impossible due to the relatively small number of infected neurons and the difficulty in imaging HSV and autophagy markers in these cells (please also refer to question 5, reviewer #1). Macrophages are not the main cells infected by HSV-1, nevertheless we tried to satisfy the reviewer's request. We infected bone marrow-derived macrophages and observed that the virus does not replicate well in these cells and there is no difference among genotypes (see data 'a' below). Moreover, colocalization of HSV-1 and LC3 in macrophages also did not show difference among groups ('b' below). These results reinforce the restrictive nature of macrophages for HSV-1 infection and the difficulty of addressing autophagy *in vivo*, where the infected cells would be limited to very specific cell types.

Figure 7 for reviewers. a, BMMs were infected with HSV-1 (KOS strain) and viral titers were measured at indicated times. **b**, Quantification of colocalization of LC3 and HSV-1 in primary macrophages. Images were analyzed by an automated pipeline created on Perkin Elmer Harmony software for colocalization quantification. Error bars are SEM. Analyzed with one-way ANOVA and Tukey post-test. ns, not significant.

5) The article of Liu et al (Cell Death Differ. 2019 Sep;26(9):1735-1749) and Gui et al

(Nature. 2019 Mar;567(7747):262-266) demonstrated that STING induces non-canonical autophagy that is dependent on ATG5. I suggest to infect ATG5 KO mice with HSV-1 to confirm a susceptible phenotype that could be linked to an autophagy protective mechanism proposed in Yamashiro et al. manuscript.

Response: We think this is a great idea. In fact, HSV-1 infection of mice harboring an ATG5 deletion in neurons was done previously by Yordy et al, 2012. The authors show that autophagy in neurons helps restrict viral replication. We don't have those mice, but we happened to have ATG5(floxed)-LysMCre mice. We infected mice via the ocular route and saw no phenotype (see below). This result was not surprising as neurons and other brain cells should still maintain autophagy competency as well as type I interferon responses.

Figure 8 for reviewers. Mice were ocular infected with 1×10^5 PFU of HSV-1 (strain 17). Viral titers in the brain at 6 days post infection. Error bars are SEM. Analyzed with one-way ANOVA and Tukey post-test. **, $p \leq 0.005$; ns, not significant.

6) Figure 4d and 4e, explain how many mice were used? Some graphs used 3 other 2 animals. The SE are big because of the few numbers of mice used to determine viral titers.

Response: We thank the reviewer for this comment. We now include the number of mice for the experiments (see Methods). To address the limited number of mice in the experiment shown, we have now plotted new data combining experiments, thus reducing the standard errors (see the revised Fig. 4d and 4e).

Reviewer #3 (Remarks to the Author):

Key findings

In the present manuscript Vance and colleagues study the importance of STING-induced type I IFN responses for antimicrobial immunity in vivo. To this end, the investigators generated two new mouse lines: one harbors a STING mutation that renders mice incapable of producing type I IFNs (S365A); another one lacks the C-terminal tail of STING, which is required for recruitment and activation of TBK1, amongst other functions. Using primary cells from these mice, the authors first validate that S365 is essential for STING-dependent type I IFN induction, but not required for promoting autophagy, whereas cells derived from the

delta CTT STING mutant mice are defective for both type I IFN induction as well as autophagy induction. The authors then move on to compare the responses of these mice strains and, as a control, the STING-null goldenticket mice, towards infection with *Mycobacterium tuberculosis* and HSV-1, respectively. Whilst there is no difference in the in vivo response towards Mtb infection, the authors show that STING S365A mice do not display an increased susceptibility towards infection with HSV-1 despite majorly compromised type I IFN induction. This result is unexpected, since the general view holds that type I IFN secretion is a major antiviral function of STING.

Overall the study is well-conducted and the results are interesting to a broader community interested in antiviral immunity and the cGAS-STING pathway. Thus, I am in support of publication in *Nature Communications*. There are a few minor comments, that the authors may want to address before publication.

One minor point relates to the interpretation of the eye infection model, which seems to support an at least partially protective function of STING-mediated cytokine secretion, a point which is not further discussed. Likewise, there appears to be more general role of S365 in regulating cytokine production (including TNF α , IL6) - at least during HSV-1 infection in vivo. Again, a minor aspect that could deserve some space in the discussion.

Response: We thank the reviewer for the generous summary of our manuscript. We appreciate that the reviewer is overall supportive of publication.

With respect to the question of whether the partially protective function of STING might be via NF- κ B-induced cytokines, we agree with the reviewer that this is a possibility. It is difficult to rule out entirely. However, when we analyzed TNF- α expression in the brain stems of mice using the eye infection model, we didn't observe dramatic differences across the various genotypes (see below). This is different than with systemic infection and suggests that perhaps STING-induced cytokines are not playing a critical role. These new data are included as supplemental Figure S4c. We don't want to draw much of a conclusion from these data since a small amount of cytokine might be playing an important protective role, but we feel it is useful to include, and we thank the reviewer for the suggestion.

Figure 9 for reviewers (and Fig. S4c of the revised manuscript). Mice were ocular infected with HSV-1 (strain 17) and brain stems collected at day 3. Relative expression of TNF- α mRNA was measured from brain stems. Error bars are SEM. Analyzed with one-way ANOVA and Tukey post-test. ns, not significant.

The main hypothesis put forward by the authors is that STING regulates virus infection through autophagy. With all the mutant mice at hand, one additional experiment that could be added to strengthen this conclusion was to infect BMDMs in vitro with HSV-1 and assess

autophagy markers - as performed in Fig. 1 for cGAMP - also, since the authors criticize “artificial in vitro stimulation” for validating STING-dependent autophagy.

Response: Macrophages are not cells that support infection or replication of HSV-1. Nevertheless, we tried the experiment that the reviewer suggests. However, we were unable to see a phenotype in these cells (Fig. 7 for reviewers; also, please refer to question 4, reviewer #2).

Reviewer #1 (Remarks to the Author):

The authors have made an attempt to address the points raised, and some of the data added have improved the manuscript. However, I find that the most critical points remain unresolved.

For instance the lack of demonstration of STING-dependent autophagy-dependent antiviral activity is a critical weakness. The authors base part of their argument on the Yordy paper, where autophagy in neurons is demonstrated to exert anti-HSV activity. However, neurons do not express STING. It is hence not likely that the autophagy response shown to be antiviral is activated through STING.

In addition, regarding the TBK1 dependence, the authors argue that it is the previous finding or TBK1-independent STING-driven autophagy by Giu et al that is surprising. However, here the authors ignore two other papers (Cell Death Differ. 2019 Sep;26(9):1735-1749; EMBO J. 2018 Apr 13;37(8). pii: e97858) also showing TBK1-independent activation of autophagy by STING. Therefore, the authors need to mechanistically explain their observation.

Finally, the preliminary data on IRF3/7 DKO mice should be consolidated to a conclusive level and integrated into the paper

Reviewer #2 (Remarks to the Author):

The authors responded my critiques based on new experiments performed. Therefore, I am satisfied with the manuscript revised version. The findings are novel and will open a new avenue to define the STING-mediated protection to HSV infection independent of type I IFN.

Reviewer #3 (Remarks to the Author):

The authors responded appropriately to my concerns. It is an interesting study that should be published.

REVIEWERS' COMMENTS:

Reviewer #1 (Remarks to the Author):

The authors have made an attempt to address the points raised, and some of the data added have improved the manuscript. However, I find that the most critical points remain unresolved.

For instance the lack of demonstration of STING-dependent autophagy-dependent antiviral activity is a critical weakness. The authors base part of their argument on the Yordy paper, where autophagy in neurons is demonstrated to exert anti-HSV activity. However, neurons do not express STING. It is hence not likely that the autophagy response shown to be antiviral is activated through STING.

Response: We appreciate the excellent point of the reviewer. We feel it is important to cite the Yordy et al paper, since it is one of the few examining the role of autophagy in HSV-1 in vivo. However, we do not claim that the relevant STING-induced autophagy response is a cell-autonomous anti-viral response that occurs in neurons. In fact, we speculate that STING-induced autophagy may be important for induction of T cell responses (e.g., by promoting antigen-presentation in APCs). Moreover, although we make the important observation that STING S365A mice can still induce autophagy, and therefore autophagy is a reasonable candidate for mediating protection to HSV-1, we leave open the possibility that other STING-dependent processes are important for mediating resistance to HSV-1. At this point it is impossible for us or anyone to address this further since there is no way to specifically eliminate STING-induced autophagy (without affecting other processes). Thus, we consider the main contribution of our manuscript to be the very surprising finding that STING-S365-induced IFN is not essential for protection against HSV-1. We think this finding will be built upon in future studies to further address the mechanism and importance of autophagy-induction (or perhaps other pathways) by STING.

In addition, regarding the TBK1 dependence, the authors argue that it is the previous finding of TBK1-independent STING-driven autophagy by Giu et al that is surprising. However, here the authors ignore two other papers (Cell Death Differ. 2019 Sep;26(9):1735-1749; EMBO J. 2018 Apr 13;37(8). pii: e97858) also showing TBK1-independent activation of autophagy by STING. Therefore, the authors need to mechanistically explain their observation.

Response: Thanks for allowing us to address these points. First, we would like to mention that we did not ignore the two papers mentioned by the reviewer; in fact they were both cited in the text of our manuscript. As we also pointed out in the manuscript, the conclusion of TBK1-independent autophagy induction in the Liu et al paper is based on experiments in immortalized HeLa cells using TBK1 knockdowns (rather than true knockouts). Therefore, we do not feel this paper impacts our conclusion, which is based on studies of primary cells with mutations made in the endogenously expressed gene. As for the Prabakaran et al paper in EMBO J (2018), we are confused by the claim of the reviewer. In our view, the conclusion of this paper is captured by its title: "Attenuation of cGAS-STING signaling is mediated by a p62/SQSTM1-dependent autophagy pathway activated by TBK1". It seems clear that this paper is claiming that autophagy induction downstream of STING depends on TBK1 and does not in fact provide evidence of STING-induced TBK1-independent autophagy (as claimed by the reviewer).

Finally, the preliminary data on IRF3/7 DKO mice should be consolidated to a conclusive level and integrated into the paper

Response: Unfortunately, our experiments with IRF3/7 DKO mice failed to yield consistent results. We are reluctant to publish these results as we do not feel they present a clear conclusion. The major problem with these experiments, as we discussed previously, is that IRF3/7 are required for IFN induction downstream of other receptors known to be important for resistance to HSV-1 (e.g., TLR3). Thus, it is difficult to reach any conclusion as to the importance of STING-induced IRF7. That is, in fact, why our generation of S365A mice is important, as this is the only way to selectively eliminate STING-induced IFN without affecting IFN induction via other receptors. To address the reviewer's concern, we now formally acknowledge a possible role for IRF7 in the discussion.

Reviewer #2 (Remarks to the Author):

The authors responded my critiques based on new experiments performed. Therefore, I am satisfied with the manuscript revised version. The findings are novel and will open a new avenue to define the STING-mediated protection to HSV infection independent of type I IFN.

Reviewer #3 (Remarks to the Author):

The authors responded appropriately to my concerns. It is an interesting study that should be published.

Response: We thank reviewers #2 and #3 for their evaluation and contributions to improve the manuscript.